# Multidimensional scaling informed by *F*-statistic: Visualizing grouped microbiome data with inference

**Hyungseok Kim**[1,2][¶][¤a], **Soobin Kim**[3][¶], **Jeffrey A. Kimbrel**[4], **Megan M. Morris**[4][¤b], **Xavier Mayali**[4], **Cullen R. Buie**[1,5]*

**1** Department of Mechanical Engineering, Massachusetts Institute of Technology, Cambridge, Massachusetts, United States of America, **2** Institute for Data, Systems, and Society, Massachusetts Institute of Technology, Cambridge, Massachusetts, United States of America, **3** Department of Statistics, University of California, Davis, Davis, California, United States of America, **4** Physical and Life Sciences Directorate, Lawrence Livermore National Laboratory, Livermore, California, United States of America, **5** Department of Biological Engineering, Massachusetts Institute of Technology, Cambridge, Massachusetts, United States of America

¶ These authors also contributed equally to this work.
¤a Current address: Global Manufacturing Sciences and Technology, Sanofi, Framingham, Massachusetts, USA
¤b Current address: Oak Ridge Institute for Science and Education, Department of Energy, Oak Ridge, Tennessee, USA
* crb@mit.edu

## Abstract

Multidimensional scaling (MDS) is a widely used dimensionality reduction technique in microbial ecology data analysis that captures the multivariate structure of the data while preserving pairwise distances between samples. While improvements in MDS have enhanced the ability to reveal group-specific data patterns, these MDS-based methods require prior assumptions for inference, limiting their application in general microbiome analysis. In this study, we introduce a new MDS-based ordination method, "*F*-informed MDS," which configures the data distribution based on the *F*-statistic, the ratio of dispersion between groups sharing common and different characteristics. Using semisynthetic datasets, we demonstrate that the proposed method is robust to hyperparameter selection while maintaining statistical significance throughout the ordination process. Various quality metrics for evaluating dimensionality reduction confirm that *F*-informed MDS is comparable to state-of-the-art methods in preserving both local and global data structures. Its application to a diatom-associated bacterial community suggests the role of this new method in interpreting the community's response to the host. Our approach offers a well-founded refinement of MDS that aligns with statistical test results, which can be beneficial for broader multidimensional data analyses in microbiology and ecology. This new visualization tool can be incorporated into standard microbiome data analyses.

**Data availability statement:** All methods and data used are publicly available on Dryad at https://doi.org/10.5061/dryad.vmcvdnd3x. An R package implementation is available at https://bioconductor.org/packages/FinfoMDS/.

**Funding:** This work was supported by U.S. Department of Energy's Biological Systems Science Division's microBiospheres Scientific Focus Area grant SCW1039 (H.K., J.A.K., M.M.M., X.M., C.R.B.). Part of this work performed under the auspices of the U.S. Department of Energy by Lawrence Livermore National Laboratory under Contract DE-AC52-07NA27344. Funders did not play any role in study design, data collection and analysis, decision to publish, or preparation of the manuscript.

**Competing interests:** The authors have declared that no competing interests exist.

## Author summary

Multidimensional scaling (MDS), also known as principal coordinate analysis, is a fundamental step in exploratory data analysis for interpreting microbial community samples processed via high-throughput sequencing. The interpretation of MDS results often involves linking patterns obtained from MDS with experimental treatments applied to the samples, such as environmental conditions or host phenotypes. However, retaining these patterns during ordination is not always guaranteed, as MDS itself does not consider group information during its learning process. This limitation reduces the effectiveness of conventional MDS, particularly for general microbiome datasets, where maintaining meaningful biological patterns is crucial. To address this gap, we present a robust statistical framework designed to represent microbiome datasets in a lower dimension, while preserving hypothesis testing results for group differences in the original dimension. Our approach, which relies on sample dispersion measured by the *F*-statistic, ensures a more stable and reliable performance compared to existing ordination methods. By incorporating statistical rigor into the ordination process, our framework improves the visualization of microbial community data and allows configurations to be adjusted within reasonable limits. This advancement provides researchers with a more effective tool for analyzing and interpreting complex microbiome data, ultimately leading to insightful conclusions.

## Introduction

Understanding microbial diversity has been advanced by the development of gene sequencing technology and multivariate data analytics, which together attempt to interpret the composition of microbial communities by targeting a conserved gene region in this phylogeny (16S rRNA gene) or by profiling all genes present (i.e., whole metagenome sequencing). Often referred to as the microbiome, the ecological structures obtained from these sequencing tools are distinguished from other biological data types in that they are compositional [1,2], sparse [3], and that their features are linked under a phylogenetic tree, providing additional genomic context [4,5]. Exploratory analysis of microbiome data [5,6], after preprocessing the sequencing reads with data normalization [7–10], typically begins with visualizing the multivariate structure to identify data patterns, distinguish variations, or remove outliers. Heatmaps or bar plots can directly represent the relative abundances of each taxon or feature. However, for large datasets including high-throughput sequencing data, statistical ordination analysis is often required to detect patterns.

The most commonly used ordination method in ecology [11] is multidimensional scaling (MDS), an ordination technique that represents data in a lower-dimensional space (e.g., two-dimensional or 2D) while preserving its original distance structure (Fig 1A). This is achieved by minimizing a stress function [12], defined as the summation of all pairwise distances or dissimilarities. Using such a dissimilarity metric is

**A** Microbial ecology data interpretation

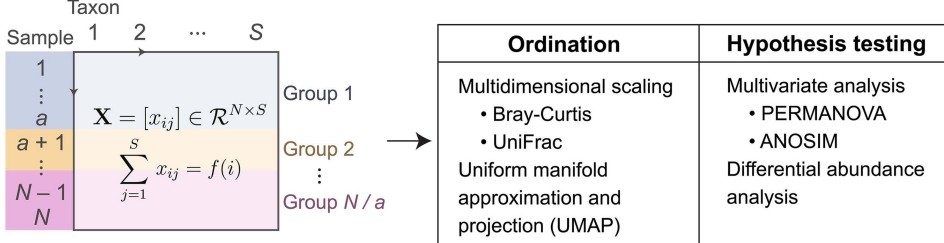

**B** Workflow of *F*-informed MDS

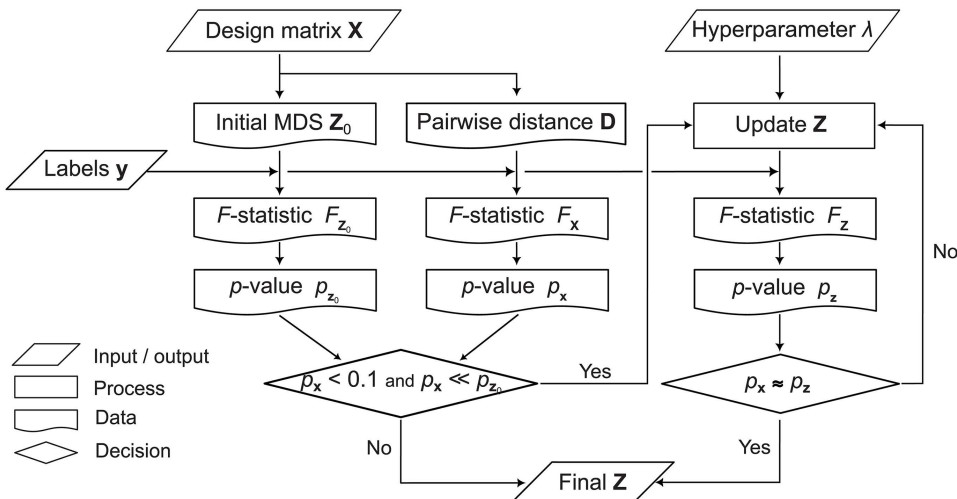

**Fig 1. *F*-informed multidimensional scaling (MDS) for microbial ecology data analysis. (a)** Schematic overview of analyzing a dataset, represented as a design matrix $\mathbf{X}$ of $N$ samples with $S$ features or taxa. In a balanced design, samples are grouped by $a$ replicates of the same experimental condition. Note that compositionality requires that the summation of elements $x_{ij}$ across features is fixed or independent of a feature $j$ (e.g., 1). Exploratory analysis of $\mathbf{X}$ is performed through its ordination, followed by statistical inference, including hypothesis testing (e.g., PERMANOVA) or differential abundance analysis. **(b)** Computational process diagram of reducing the dimensionality of $\mathbf{X}$ using label set $\mathbf{y}$ for visualization, ensuring compliance with inference results based on the *F*-statistic.

essential for finding an appropriate 2D representation of these sparse or zero-inflated compositions [13], and different patterns have been observed depending on the choice of metric [5,14]. Examples in microbiome analysis include Bray-Curtis [15] and Unifrac [16,17], which incorporate 16S rRNA gene-based taxonomic information as well as the compositional structure. Herein, we refer to these distance-based ordination methods to as metric MDS. Applications of distance metrics have allowed MDS-based ordination to detect patterns and environmental gradients [18], or further address biologically confounding factors [19,20]. More recently, an alternative visualization tool for high-dimensional biological data, UMAP [21], has also been applied to microbiomes to cluster features associated with their sampling site within a host [22].

Evaluating ordination or dimension reduction methods has been facilitated by establishing quantitative measures to assess their performance or appropriateness for specific objectives. These evaluations quantify the change or loss of information from the original structure using quality metrics widely employed in information visualization [23–25], with applications in biological fields such as single-cell genomics [26] and transcriptomics [27,28]. These metrics assess how well local patterns are preserved (e.g., trustworthiness, continuity [29]) or how few pairwise distances are distorted from the original structure (e.g., Shepard diagram, normalized stress [30]). For microbiome data, however, the evaluations have

primarily focused on the ability to identify group clustering on a case-by-case basis, using measures such as Rand index [14], k-means clustering [18,20], and the $F$-statistic [20,31]. Additionally, environmental gradient patterns have been analyzed using kernel- [32] or rank-based [18] regressions.

Notably, most ordination methods for ecological data analysis do not utilize data labels and instead rely on unlabeled or unweighted distances in their computations. To test for structural change and to correlate results with treatment effects, an additional and independent differential abundance analysis is required [5,6]. While metric MDS can detect some changes [33], its low-dimensional representation does not always capture every underlying pattern—even when statistical validation confirms the presence of such a pattern in the original space. A pattern is pronounced in the low-dimensional embedding only if the distance structures between groups are sufficiently distinct to allow a clear visualization. When this is not the case, scree plot analysis can be used to determine the dimensionality at which the pattern becomes visible. However, a scree plot does not indicate how the samples are distributed within this reduced dimension. Moreover, if the required dimensionality exceeds three, metric MDS cannot effectively visualize the difference between groups.

This limitation has driven the development of MDS-based methods that incorporate additional contextual information to improve pattern detection. One approach, broadly referred to as confirmatory MDS [34,35], differentiates between sample groups by incorporating labels [36,37]. This method is enabled by hyperparameter adjustments that optimize the objective function [20,37,38]. Adapted representations are generally accepted as long as they do not deviate significantly from the metric MDS representation [12], particularly given that MDS optimizes a non-convex function and can yield different solutions [39,40]. For biological data applications, newer visualization methods have been introduced to address confounding factors by making certain assumptions about the relationship between errors and covariates [19] or by incorporating group labels into the latent structure [20,41]. However, applying these methods to general microbiome datasets is limited by the assumptions regarding structural differences or data models, which are not always valid in real-world scenarios.

In this work, we propose a novel framework, $F$-informed MDS ($F$-MDS), designed to visualize statistical differences within an exploratory MDS setting by leveraging the $F$-statistic (Fig 1B). Our approach aims to introduce minimal perturbations to the metric MDS representation, preserving the global distance structure as much as possible. We applied the perturbation only to datasets for which metric MDS failed to detect a statistically significant group-wise difference under the original data structure. This generalizes previous label-based and model-based MDS methods by eliminating the need for prior assumptions about group structures or their differences. In the following sections, we demonstrate how $F$-informed MDS produces representations that are less dependent on hyperparameter choices and evaluate its performance in comparison to other dimension reduction tools.

## Methods

In this section, we review the terminology and definitions of multidimensional scaling (MDS), along with an example of its supervised variant that incorporates sample group information. Next, we propose a new MDS method and discuss multivariate analysis of variance via permutation (i.e., PERMANOVA [42]). Finally, we introduce and justify quality metrics for evaluating dimension reduction tools and compare them with the proposed MDS approach.

### Review of multidimensional scaling methods

We consider a balanced design where an $i$-th sample, denoted as $\mathbf{x}_i \in \mathbb{R}^S$, is $S$-dimensional and associated to a discrete label $y_i \in \{1, 2, \cdots, G\}$, representing $G$ groups, for every index $i = 1, 2, \cdots, N$ (the number of samples, Fig 1A). The pairwise distance between $\mathbf{x}_i$ and $\mathbf{x}_j$, denoted $d_{ij} \in \mathbb{R}_+$, is computed using an arbitrary dissimilarity metric, such as Euclidean, Bray-Curtis [15], or UniFrac [16]. Let $\mathbf{D} = [d_{ij}]$ denote the matrix of the pairwise distances.

In the metric multidimensional scaling, the goal is to find a two-dimensional representation $\mathbf{Z} = (\mathbf{z}_1, \cdots \mathbf{z}_N) \in \mathbb{R}^{2 \times N}$ that best preserves the original distance structure $\mathbf{D}$. This is enabled by minimizing an objective function $O_{\text{MDS}}(\mathbf{Z})$ (termed the raw stress [30]),

$$O_{\text{MDS}}(\mathbf{Z}) = \frac{1}{2} \sum_{i,j} (d_{ij} - \|\mathbf{z}_i - \mathbf{z}_j\|_2)^2.$$

(1)

Known algorithms to minimize the raw stress include matrix decomposition via Nyström approximation [43], divide-and-conquer [44,45] for Euclidean distances, or more generally for non-convex functions, greedy algorithms [39] such as majorization [12,46] or stochastic gradient descent [40,47,48].

While the metric MDS is an unsupervised learning and does not require the label set $\mathbf{y} \in \{0, 1\}^N$, supervised versions of multidimensional scaling such as SuperMDS [38], impose additional constraints on the representation by class groups. The purpose is to classify samples by the labels as well as to compute the multidimensional scaling. This is enabled by adding a confirmatory or constraining term to the raw stress,

$$O_{\text{SMDS}}(\mathbf{Z}) =$$
$$(1 - \alpha) \cdot \underbrace{\frac{1}{2} \sum_{i,j} (d_{ij} - \|\mathbf{z}_i - \mathbf{z}_j\|_2)^2}_{\text{raw stress}} + \alpha \underbrace{\sum_{i,j:y_j>y_i} (y_j - y_i) \sum_{k=1}^{2} \left( \frac{d_{ij}}{\sqrt{2}} - (z_{jk} - z_{ik}) \right)^2}_{\text{confirmatory term}},$$

(2)

where the confirmatory term involves the group labels. The two terms are balanced by a hyperparameter $\alpha \in [0, 1]$ that controls the degree of classification. Minimizing $O_{\text{SMDS}}(\mathbf{Z})$ can locate the representation points closer to each other when they are within the same group. Selection of the hyperparameter is carefully guided by the data structure, so that the process avoids unnecessary group distinctions during the visualization.

## Proposal of multidimensional scaling informed by pseudo *F*-statistic

Here, we introduce a test statistic for multivariate data based on sample dispersion (i.e., *F*-statistic) and propose a revised MDS framework that utilizes this statistic.

**Permutational multivariate analysis of variance.** Testing for group differences in multivariate data is commonly performed by calculating the *F*-statistic, which compares inter- and intra-group variances based on pairwise distances. However, a standard *F*-test assumes that observations are normally distributed with equal variance, an assumption that does not hold for most compositional or zero-inflated data. Instead, a "pseudo" *F*-statistic has been defined [49], by denoting $\epsilon_{ij}$ the indicator function $\mathbb{1}\{y_i = y_j\}$ to write

$$F = \left( \frac{\sum_{i,j} d_{ij}^2}{G \sum_{i,j} \epsilon_{ij} d_{ij}^2} - 1 \right) \cdot \frac{N - G}{G - 1}.$$

(3)

The empirical distribution of the pseudo-*F* statistic is approximated by label permutation for sufficiently large datasets [13,50]. Specifically, $K$ independent permutations $\Pi_1, \cdots, \Pi_K$ generate permuted label sets $[y_i^{\Pi_k}]_{i=1}^N$, from which the values $F^{\Pi_k}$ are computed. This pseudo-*F*-distribution is then used to obtain a *p*-value, a procedure known as permutational multivariate analysis of variance (PERMANOVA) [42]:

$$p = \frac{1}{K} \sum_{k=1}^{K} \mathbb{1}\{F^{\Pi_k} \geq F\}.$$

(4)

**Proposal of *F*-informed MDS.** To incorporate hypothesis testing into MDS analysis, we developed a weakly supervised learning method (Fig 1B). First, a two-dimensional representation was initialized by computing metric MDS

(i.e., an unsupervised step). Next, the representation points were iteratively adjusted so that the $p$-value calculated in 2D space matches to the original PERMANOVA $p$-value. This was achieved by defining an objective function in which a new confirmatory term was added to the raw stress,

$$O_{\text{FMDS}}(\mathbf{Z}) \sim \underbrace{\frac{1}{2}\sum_{i,j}(d_{ij} - \|\mathbf{z}_i - \mathbf{z}_j\|_2)^2}_{\text{raw stress}} + \lambda \cdot \underbrace{\left|F_\mathbf{z} - f_\mathbf{z}(F_\mathbf{x})\right|}_{\text{confirmatory term}},$$

(5)

where the confirmatory term minimized the discrepancy between the $p$-values from the permutation $F$-distributions obtained in each space, namely the original and the two-dimensional (denoted $F_\mathbf{x}$ and $F_\mathbf{z}$, respectively). Because the two $F$-ratios were scaled differently, we introduced a local regression function $f_\mathbf{z}$ to map between them (Algorithm 1). In addition, prior to the regression, each ratio was computed iteratively using a set of permuted labels $\mathbf{y}^{\Pi_k}$ for $k = 1, \cdots, K$, and the resulting values were sorted by magnitude. This enabled the original $F$-ratio (from the unpermuted labels) to be mapped precisely in the two-dimensional space. Details of the mapping function and the derivation of the exact $O_{\text{FMDS}}$ are given in Appendix A, S1 File.

### Algorithm 1 Regression function $f_\mathbf{z}$ for mapping between $F_\mathbf{x}$ and $F_\mathbf{z}$.

```
         z ∈ ℝ^{2×N}: 2D representation
Require: D ∈ ℝ_+^{N×N}: pairwise distance
         y = {1, 2, ⋯ G}^N: labels or groups
Ensure:  f_z : ℝ → ℝ
 1: ℒ_x ⇐ empty list
 2: ℒ_z ⇐ empty list
 3: for 1 ≤ k ≤ 999 do
 4:    y^Π ⇐ random permutation on y
 5:    F_x^Π ⇐ pseudo-F using D and y^Π
 6:    F_z^Π ⇐ pseudo-F using Z and y^Π
 7:    ℒ_x ⇐ append F_x^Π to ℒ_x
 8:    ℒ_z ⇐ append F_z^Π to ℒ_z
 9: end for
10: ℒ_x ⇐ sort ℒ_x by an increasing order
11: ℒ_z ⇐ sort ℒ_z by an increasing order
12: f_z ⇐ local regression between ℒ_x and ℒ_z
```

**Majorization algorithm.** We implemented the majorization (or Majorization-Minimization) algorithm to minimize the $F$-MDS objective (Eq 5). First, an exact form of $O_{\text{FMDS}}(\mathbf{Z})$ was derived to match the physical dimension of each term (Appendix A, S1 File),

$$O_{\text{FMDS}}(\mathbf{Z}) = \frac{1}{N}\sum_{i,j}(d_{ij} - \|\mathbf{z}_i - \mathbf{z}_j\|_2)^2 + \lambda \left|\sum_{i,j}\left[1 - G\epsilon_{ij}\left(1 + \frac{G-1}{N-G}f_\mathbf{z}(F_\mathbf{x})\right)\right]\|\mathbf{z}_i - \mathbf{z}_j\|_2^2\right|.$$

(6)

Next, as similarly described in [30,38], the majorization was applied to seek an optimal $k$-th point for every $k = 1, \cdots N$, while keeping other points fixed:

$$\mathbf{z}_k^* = arg\,min_{\mathbf{z}_k} O_{\text{FMDS}}(\mathbf{Z}|\mathbf{z}_1, \cdots \mathbf{z}_{k-1}, \mathbf{z}_{k+1}, \mathbf{z}_N).$$

(7)

It can be analytically shown that Eq 7 can be majorized with a quadratic expression in $\mathbf{z}_k$, which is further minimized by taking a derivative with respect to each element of $\mathbf{z}_k$ (denoted as $z_{ks}$, $s = 1,2$). An update rule was established by solving for

$z_{ks}$ after setting the derivative to zero. It should be noted that the update is applied exclusively to the representation where its group-wise difference is significant under the original structure ($p_{\mathbf{x}} < 0.1$), but not identified in the two-dimensional embedding ($p_{\mathbf{z}} \gg p_{\mathbf{x}}$). The update was designed to continue until when the difference between $p_{\mathbf{x}}$ and $p_{\mathbf{z}}$ is minimized. Detailed derivations of the expression and the update rule for $\mathbf{Z}$ are described in Appendix B, S1 File, which is summarized in Algorithm 2.

## Algorithm 2 Update rule for computing *F*-MDS.

```
          λ ∈ ℝ₊: hyperparameter
Require:  D ∈ ℝ₊^{N×N}: pairwise distance
          y = {1, 2, ⋯ G}^N: labels or groups
Ensure:   Z ∈ ℝ^{2×N}: 2D representation
1:  Fₓ ⇐ pseudo-F using D, y (Eq 3)
2:  p_{x, new} ⇐ p-value for group difference in X
3:  p_{z, new}, p_{z, old} ⇐ p-value for group difference in Z
4:  while (|p_{z, new} − pₓ| > 0.05) ∨ (|p_{z, new} − pₓ| < |p_{z, old} − pₓ|) do
5:    p_{z, old} ⇐ p-value for group difference in Z
6:    for 1 ≤ k ≤ N do
7:      δ(Z) ⇐ sign of confirmatory term (Equation S13)
8:      f_z(Fₓ) ⇐ mapping function (Algorithm 1)
9:      z_k ⇐ update z_k using δ(Z) and f_z(Fₓ) (Equation S16)
10:   end for
11:   p_{z, new} ⇐ p-value for group difference in Z
12: end while
```

## Dataset and evaluation

**Semisynthetic data.** The behavior of *F*-informed MDS is illustrated using simulation studies with semisynthetic datasets designed to mimic a microbiome community. Among eleven available simulation packages [51], SparseDOSSA [52] was chosen for its ability to capture key features of ecological structure including compositionality, sparsity, and feature-wise correlation. SparseDOSSA is a Bayesian model where the sequencing count for a microbial gene feature $s \in \{1, \cdots S\}$ follows a multinomial distribution with probabilities equal to the corresponding relative abundances of the features. The relative abundances are obtained by normalizing absolute abundances (denoted as $a_s$), and each absolute abundance marginally follows a zero-inflated log-normal distribution. Specifically, the marginal specification of $a_s$ is

$$a_s = 0 \qquad \text{with probability } \pi_s,$$
$$\log a_s \sim \mathcal{N}(\mu_s, \sigma_s^2) \text{ with probability } 1 - \pi_s, \tag{8}$$

where $\pi_s$ is the probability that feature $s$ is absent, and $\mu_s$ and $\sigma_s$ are the mean and standard deviation of the log-transformed, nonzero feature abundance, respectively. In addition, dependency among features in SparseDOSSA is indirectly handled by the copula parameter $\Omega \in \mathcal{R}^{S \times S}$. This parameter describes a latent multivariate Gaussian variable that is compared with $\pi_s$ to determine whether the feature is present.

In the simulation studies, SparseDOSSA parameters were estimated by fitting the model to shotgun-sequenced metagenomes of healthy human stool ("stool" [53]), followed by mean adjustments. The training set consists of 239 stool samples, each comprising 332 microbial features. For the binary-group simulation (i.e., $y = 1, 2$), two balanced semisynthetic datasets were generated using the fitted stool parameters, with all but one feature mean differently adjusted for each dataset. Specifically, a feature $s_1$ with the smallest "effective" variance $\sigma_{\text{eff}}^2$ was selected (Eq 9); an offset value of $\pm 5$ for each group was then added to that feature's mean, i.e., $\mu_s(s = s_1)$ (Eq 10).

$$\sigma_{\text{eff}}(a_s) \triangleq \sqrt{(1 - \pi_s) \cdot \text{Var}(a_s) + \pi_s(1 - \pi_s) \cdot \mathbf{E}(a_s)^2},$$
$$\text{Var}(a_s) \triangleq e^{2\mu_s + \sigma_s^2}(e^{\sigma_s^2} - 1), \quad \mathbf{E}(a_s) \triangleq e^{\mu_s + \sigma_s^2/2} \tag{9}$$

$$\text{Find}: s_1 = \arg\min_{s \in [S]} \sigma_{\text{eff}}(a_s),$$
$$\text{Adjust}: \mu_s(s_1, y) \leftarrow \mu_s(s_1, \texttt{stool}) + 5 \cdot (-1)^y, \ \forall y \in \{1, 2\} \tag{10}$$

Similarly, for ternary-group simulation ($y = 1,2,3$), three balanced semisynthetic datasets were generated using the fitted SparseDOSSA parameters after differently adjusting the mean values of five features ($s_i$, $i = 1, \cdots, 5$) (Eq 11). Again, the features were selected from the 332 features to have the smallest effective variances $\sigma_{\text{eff}}^2$.

$$\text{Find}: s_i = \arg\min_{s \in [S] \setminus \{s_1, \cdots, s_{i-1}\}} \sigma_{\text{eff}}(a_s), \ i = 2, \cdots, 5$$
$$\text{Adjust}: \mu_s(s_i, y) \leftarrow \mu_s(s_i, \texttt{stool}) + (y-1)(6-i)(-1)^i, \ \forall i \in [5], y \in [3] \tag{11}$$

In summary, the stool-trained SparseDOSSA model was adjusted for each data group to illustrate cases where metric MDS failed to reveal group-based patterns in a two-dimensional representation. The adjustments were designed to be minimal to capture the microbiome characteristics as realistically as possible. For binary dataset, the *F*-informed MDS was evaluated using the semisynthetic datasets of sizes $N \in \{50, 100, 200, 500, 1000\}$ and a hyperparameter $\lambda \in [0, 1]$. Pairwise distances between two samples were calculated based on Bray-Curtis dissimilarity on the original structure, and Euclidean distance on 2D representations. For each setting, triplicate datasets were generated.

**Real microbiome community.** We also evaluated *F*-informed MDS using bacterial community data associated with or sampled from hosts, including a photosynthetic diatom [54] and the human gut, the latter used for studying liver cirrhosis [55] or type 2 diabetes (T2D) [56,57]. For the algal-associated microbiome, the dataset consisted 36 balanced microbial community samples cultured with or without the microbial host (alga *Phaeodactylum tricornutum*). The community comprised 72 bacterial taxa as identified by amplicon sequence variants (ASV) of the 16S rRNA gene, PCR-amplified. ASV counts were obtained through the Illumina MiSeq system and subsequently converted into relative abundances using cumulative sum scaling (CSS) [10]. A phylogeny-based metric (weighted Unifrac [17]) was used to calculate pairwise distances between samples. For the human gut microbiome, two datasets generated from shotgun metagenomic sequencing were retrieved from a publicly available repository [58], where reads were merged at the genus level [59]. Detailed reproduction and processing steps for these datasets are described in Appendix D, S1 File.

**Performance evaluation metrics.** To quantify the performance of dimension reductions, four known [25] and two new quality metrics were defined and jointly used. Trustworthiness and continuity, among the first four metrics, measure how closely neighborhood points are preserved from the original (*S*-) to a two-dimensional representation, or vice versa, defined in [29]:

$$\text{Trustworthiness} = 1 - \frac{2}{Nk(2N - 3k - 1)} \sum_{i=1}^{N} \sum_{j \in U_k(i)} (r_{ij} - k)$$
$$\text{Continuity} = 1 - \frac{2}{Nk(2N - 3k - 1)} \sum_{i=1}^{N} \sum_{j \in V_k(i)} (\hat{r}_{ij} - k), \tag{12}$$

where $k$ is the size of neighborhood of interest, $r_{ij}$ and $\hat{r}_{ij}$ respectively are the distance-based ranks of *j*-th point from *i*-th point under the original and/or two-dimensional space. Sets $U_k(i)$, $V_k(i)$ are constructed by following, $U_k = \hat{C}_k \cap C_k^c$ and $V_k = C_k \cap \hat{C}_k^c$, where $C_k(i)$ and $\hat{C}_k(i)$ are the set of $k$ points closest to *i*-th point in the original and two-dimensional space, respectively. Two different $k$ values, corresponding to $\sim$ 8% and 75% of data size, were applied to evaluate how local and global information is preserved in simulated and real datasets. Trustworthiness and Continuity were computed using an R package dreval (v.0.1.5 [60]).

Additionally, we quantified the degree of deviation of the representation **Z** from the original distance **D** by calculating two metrics; the first was the "normalized" stress or Stress-1 [30],

$$\text{Stress-1} = \frac{\sum_{i,j}(d_{ij} - \|\mathbf{z}_i - \mathbf{z}_j\|_2)^2}{\sum_{i,j}\|\mathbf{z}_i - \mathbf{z}_j\|_2^2},$$

(13)

and the second using a Shepard diagram [61] and its Pearson correlation coefficient. Both metrics provided how much distortion from the original global structure had occurred in the representation from each reduction method.

Two new metrics were also defined and used, which we termed as *F*-correlation and *F*-rank-ratio,

$$F\text{-correlation} = \text{corr}(F_{\mathbf{x}}^\pi, F_{\mathbf{z}}^\Pi)$$

$$F\text{-rank-ratio} = \frac{1 - p_{\mathbf{z}}}{1 - p_{\mathbf{x}}} = \frac{\sum_{k=1}^K \mathbb{1}\{F_{\mathbf{z}}^{\Pi_k} < F_{\mathbf{z}}\}}{\sum_{k=1}^K \mathbb{1}\{F_{\mathbf{x}}^{\Pi_k} < F_{\mathbf{x}}\}},$$

(14)

with $\text{corr}(\cdot, \cdot)$ denoting the Pearson coefficient between two random variables and superscript $\Pi$ indicating a permutation of group labels **y**.

Experiments were performed using R via RStudio either on Apple M3 chip with 16 GB of RAM, or on 128-thread IBM POWER9 CPU with 256 GB of RAM at the MIT Office of Research Computing and Data. Datasets are publicly available on Dryad at https://doi.org/10.5061/dryad.vmcvdnd3x [62].

## Results

### *F*-informed MDS is robust to a choice of hyperparameter

We first characterized the behavior of *F*-informed MDS by observing changes in the MDS representation across a range of hyperparameters. After each update of the representation **Z**, the changes were recorded by computing the objective function $O_{\text{FMDS}}(\mathbf{Z})$, and PERMANOVA *p*-value $p_{\mathbf{z}}$ from testing on **Z**. The *p*-value was chosen as a tracking measure instead of the pseudo *F*-statistic (Appendix A, S1 File). To generate data, binary, balanced datasets of sizes *N* ranging from 50 to 500 were simulated using SparseDOSSA [52], a Bayesian model of microbiome communities that captures their major features. The statistical model was trained using 239 human stool-derived community samples with 332 microbial species [53], followed by mean adjustment of one feature to reflect inter-group differences (see "Semisynthetic data" in Methods). A heatmap of a simulation result, along with its histogram, confirmed that the data were sparsely normalized (S1A and S1B Fig). A two-dimensional representation obtained using principal coordinates analysis showed that the group difference was indistinguishable (S1C Fig). Indeed, PERMANOVA testing of the metric MDS representation with Euclidean distance did not detect a statistically significant difference ($p_{\mathbf{z}} > 0.51$), whereas the original data structure did reveal a significant difference ($p_{\mathbf{x}} < 0.05$, Table 1).

**Table 1. PERMANOVA test results for group differences in semi-synthetic datasets using the original structure ($p_{\mathbf{x}}$) and the principal coordinates analysis representation ($p_{\mathbf{z}}$). The datasets were generated using SparseDOSSA [52] (section "Semisynthetic data" of Methods).**

| | N=50 | | N=100 | | N=200 | | N=500 | |
|---|---|---|---|---|---|---|---|---|
| | $p_{\mathbf{x}}$ | $p_{\mathbf{z}}$ | $p_{\mathbf{x}}$ | $p_{\mathbf{z}}$ | $p_{\mathbf{x}}$ | $p_{\mathbf{z}}$ | $p_{\mathbf{x}}$ | $p_{\mathbf{z}}$ |
| Replicate 1 | 0.029 | 0.702 | 0.071 | 0.662 | 0.027 | 0.765 | 0.034 | 0.800 |
| Replicate 2 | 0.039 | 0.607 | 0.009 | 0.510 | 0.037 | 0.865 | 0.049 | 0.568 |
| Replicate 3 | 0.041 | 0.991 | 0.027 | 0.915 | 0.026 | 0.995 | 0.047 | 0.809 |

Tracking changes in the *F*-MDS representation over training epochs indicated that a minimal hyperparameter value and a stopping rule were required for the majorization algorithm to minimize the objective function ([Fig 2A]). In semisynthetic datasets generated with SparseDOSSA, we observed a gradual decrease in $p_z$-value and $O_{FMDS}(\mathbf{Z})$ over successive epochs. When training continued without a stopping rule, $p_z$-values oscillated between 0 and 1, with a higher oscillation rate at larger $\lambda$ ([S2A Fig]). Therefore, we specified a bounded region, $|p_z - p_x| < 0.05$, as the stopping criterion for terminating the majorization algorithm. Under this rule, the algorithm stopped in fewer than 30 iterations when $\lambda \geq 0.2$ ([Fig 2B]). Repeating the experiment across triplicate simulated datasets of varying sizes confirmed that $\lambda = 0.15$ was the minimal

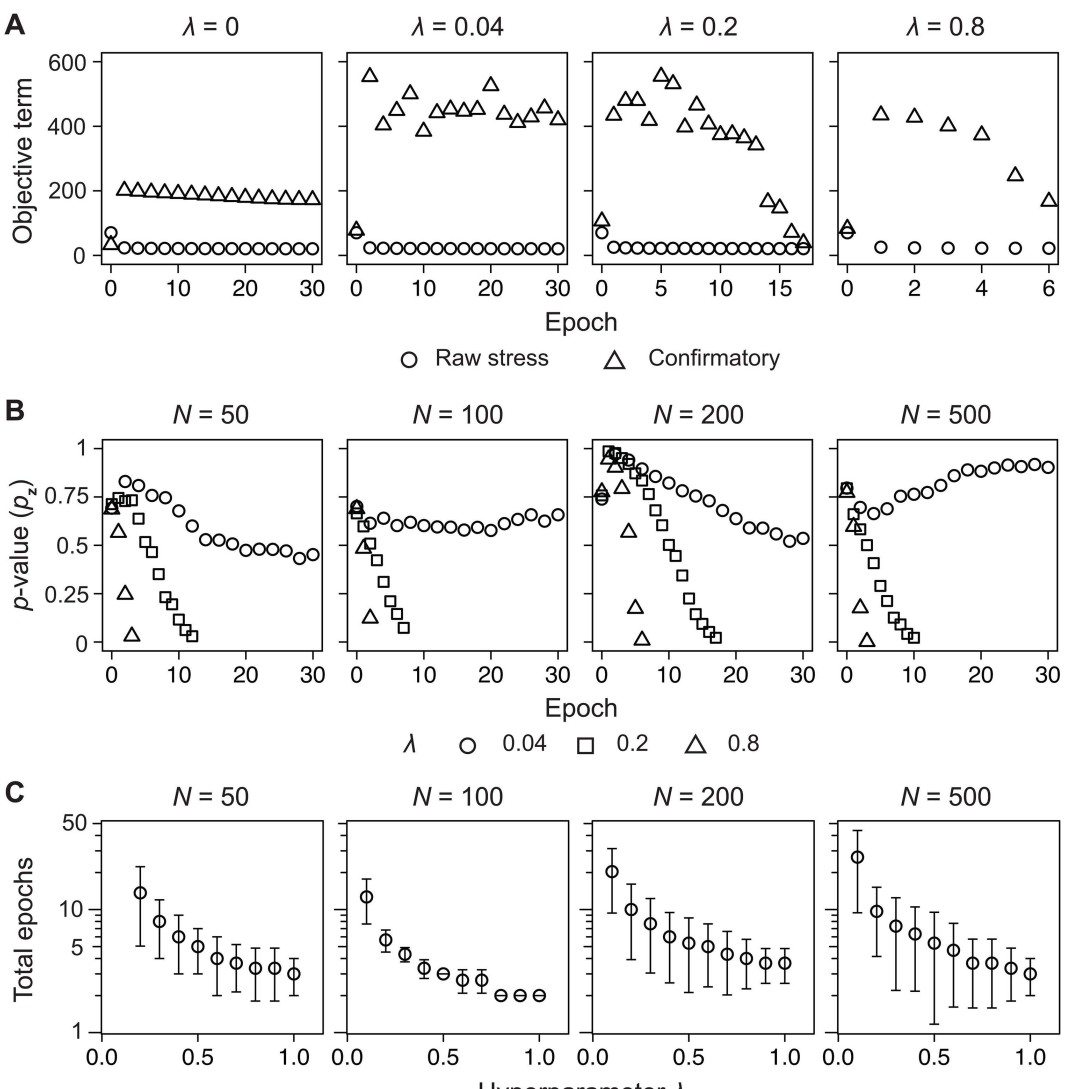

**Fig 2. Majorization algorithm optimizes the *F*-informed MDS objective function. (A)** Changes in each term comprising the objective function ([Eq 5]), including raw stress and confirmatory terms, are plotted against the training epochs for different hyperparameter values $\lambda$. A semisynthetic dataset of $N = 200$, replicate 1, was generated using SparseDOSSA [52] (see the "Semisynthetic data" section of the Methods). **(B)** The PERMANOVA *p*-value under two-dimensional representation ($p_z$) is plotted against epoch and $\lambda$ until the stopping criteria are met. **(C)** Number of epochs until the termination is plotted against the hyperparameter, ranging between 0.2 and 1. Error bars indicate the standard deviation of triplicates.

PLOS Computational Biology

value required for the termination (S2B Fig). To preserve convexity of the majorization form of $O_{FMDS}(\mathbf{Z})$ with a positive quadratic coefficient, the hyperparameter was constrained to values not exceeding unity (Fig 2C and Appendix B, S1 File).

After confirming the minimization of the objective with different hyperparameters, we next asked whether these values for $F$-MDS impacted its 2D representation. The visualizations were assessed using Shepard diagram and Stress, two quality metrics commonly used to evaluate how well an MDS representation preserves the distance structure from the original dimension (see Methods). Supervised MDS (superMDS [38]) was chosen as a benchmark ordination method because its objective function is similarly controlled by a single hyperparameter $\alpha$ within the same range [0,1], ensuring performance of its majorization algorithm. Shepard diagram analysis of these MDS methods showed that pairwise Euclidean distances deviated from the original structure in the two-dimensional representation to a greater extent when a nonzero hyperparameter was used. Interestingly, however, the change was more subtle in $F$-MDS (Figs 3A and S3A).

To further compare structural deviations, we calculated the Pearson correlation coefficient from the Shepard plot as well as the normalized stress (i.e., Stress-1, Eq 13). The pairwise distances correlation consistently remained above $\sim 0.4$ for

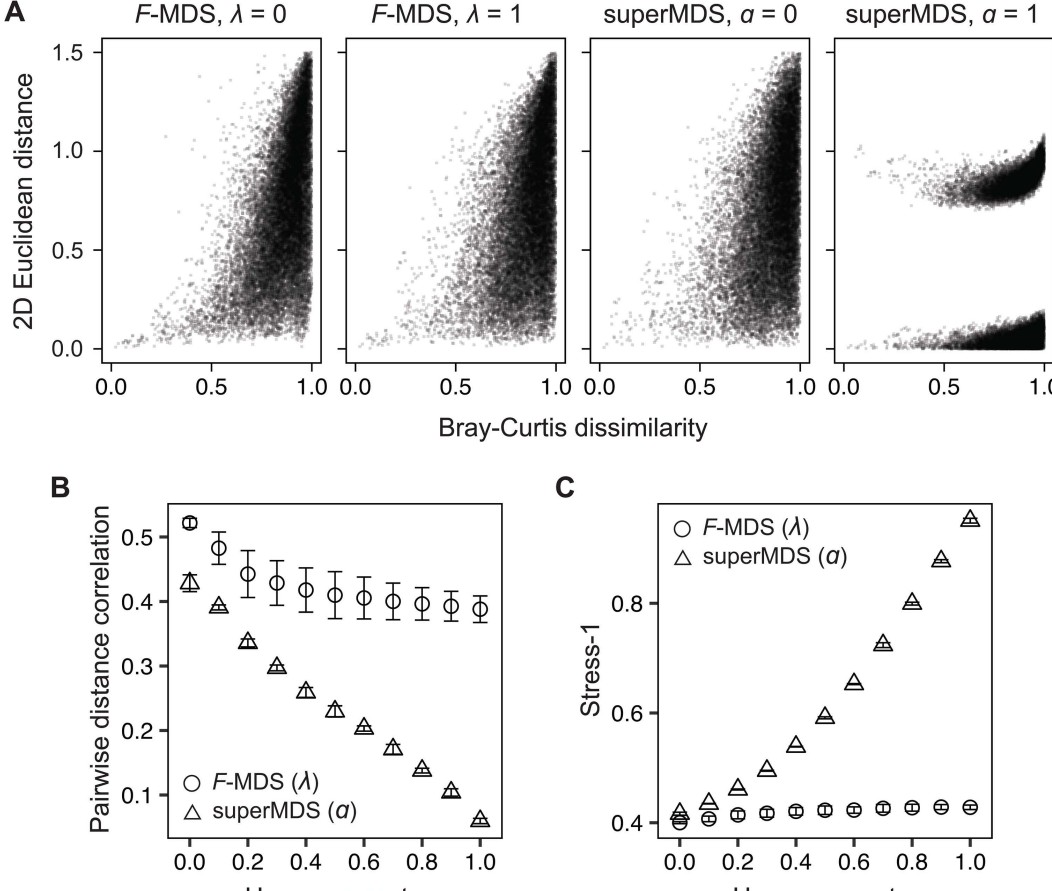

**Fig 3. $F$-informed MDS is robust to the choice of hyperparameters. (A)** Shepard plots comparing pairwise distances from $F$-informed MDS ($F$-MDS) and supervised MDS (superMDS) under both zero (metric MDS) and nonzero hyperparameter settings ($\lambda$, $\alpha$ = 1). Distances in the original high-dimensional space were computed using Bray-Curtis dissimilarity; distances in the two-dimensional embedding used Euclidean metric. **(B)** For each method, the Pearson correlation coefficient was calculated between original and embedded distances. **(C)** Normalized stress (Stress-1) was computed for each embedding. Analysis used semisynthetic datasets with $N = 200$ (see the "Semisynthetic data" section of the Methods). Error bars indicate standard deviation across triplicate datasets.

all tested values of $\lambda$ in *F*-MDS, whereas superMDS showed a decrease in correlation coefficients to below than 0.1 as $\alpha$ increased (Figs 3B and S3B). Similarly, the increase in Stress-1 due to a higher hyperparameter was 19.1-fold greater in superMDS than in *F*-MDS (Figs 3C and S3A). Together with previous results, these findings confirm that the proposed method retains the original distance information as effectively as metric MDS while being less dependent on the hyperparameter. At the same time, it successfully rearranges the 2D structure so that the groups become statistically distinct.

Based on the hyperparameter $\lambda$'s influence on the MDS configuration, a grid search was conducted to identify an optimal value. Prior analysis demonstrated a faster *F*-MDS optimization with larger $\lambda$ (Fig 2C). To balance the epoch number against structural deviation while mitigating bias across the metrics, we defined an objective function $f_{obj}(\lambda)$ via min-max normalization (Appendix C, S1 File). The loss function was evaluated over $\lambda$ between 0.15 and 1, where the performance was assured (S4A Fig). Across semisynthetic datasets, the hyperparameter $\lambda$ that minimized $f_{obj}(\lambda)$ was greater than or equal to 0.5 ($0.73 \pm 0.19$, mean ± s.d.; see S4B Fig).

**Using different quality metrics confirms the consistent performance of *F*-MDS**

Our earlier results demonstrate that *F*-informed MDS effectively represented the distance structure in 2D space. Next, we sought to generalize our statistical approach by comparing *F*-MDS to other dimension reduction methods, as recent examples have shown these methods can be also used to visualize microbiome data. To evaluate each ordination more comprehensively, we introduced four quality metrics alongside those previously discussed (i.e., pairwise distance correlation and Stress-1, see "Performance evaluation metrics" in Methods). The first two, Trustworthiness and Continuity [29], measure the degree to which local neighborhood structures are preserved in the 2D representation (Eq 12). While these metrics do not require group labels, they have been used in previous studies to evaluate gene expression analysis methods [63,64]. The other two metrics assess how closely pseudo *F*-distributions derived from the original multidimensional structure are reflected in the 2D representations (Eq 14). First, *F*-correlation measures the Pearson correlation between two *F*-ratios: one computed from the original data's distance structure and the other from 2D representation using permuted labels, as described in [65]. Each pair of *F*-ratios was obtained from single permutation, and $K = 500$ permutations were performed to compute the *F*-correlation coefficient. A high *F*-correlation value indicates that the ordination method successfully reconstructs a similar dispersion pattern in the lower-dimensional space. Second, the *F*-rank-ratio was defined as the ratio of the reversed rank of *F* from unpermuted labels to the rank of an ordered *F*-set obtained from *K* permutations. This ratio was calculated for both the original and 2D representations. An *F*-rank-ratio close to 1 indicates that the ordination accurately preserves the statistical significance of group differences as presented in the original structure. These two metrics were used together to evaluate whether an ordination method retains the original pseudo *F*-distribution while also preserving statistical differences between groups.

**Semisynthetic dataset.** Using six quality metrics, the performance of *F*-MDS was evaluated and compared to other ordinations, including metric MDS, UMAP [21], t-SNE [66], and Isomap [67]. Semisynthetic dataset was first used to compute the metrics across a range of hyperparameters. We observed that for local neighborhoods preservation, i.e., 7% of data size, unsupervised UMAP (UMAP-U) achieved the highest trustworthiness score, followed by t-SNE, supervised UMAP, (*F*-)MDS, and Isomap (Figs 4A and S5A). While both unsupervised and supervised UMAP exhibited higher trustworthiness (>0.73), the latter showed lower continuity, suggesting that additional constraints were introduced into the 2D representation when group labels were incorporated. In contrast, *F*-MDS achieved a relatively high continuity of $\sim$0.77 while consistently maintaining a trustworthiness score $\sim$0.73. Overall, UMAP-U best preserved local neighborhood information for these simulated data, while our method demonstrated comparable performance with lower reliance on the hyperparameter.

For global structure preservation, MDS-based ordinations (e.g., metric MDS, *F*-MDS, Isomap) were the top performers, as expected, when evaluated using the same two metrics but with broader neighborhoods (75% data size, Figs 4B

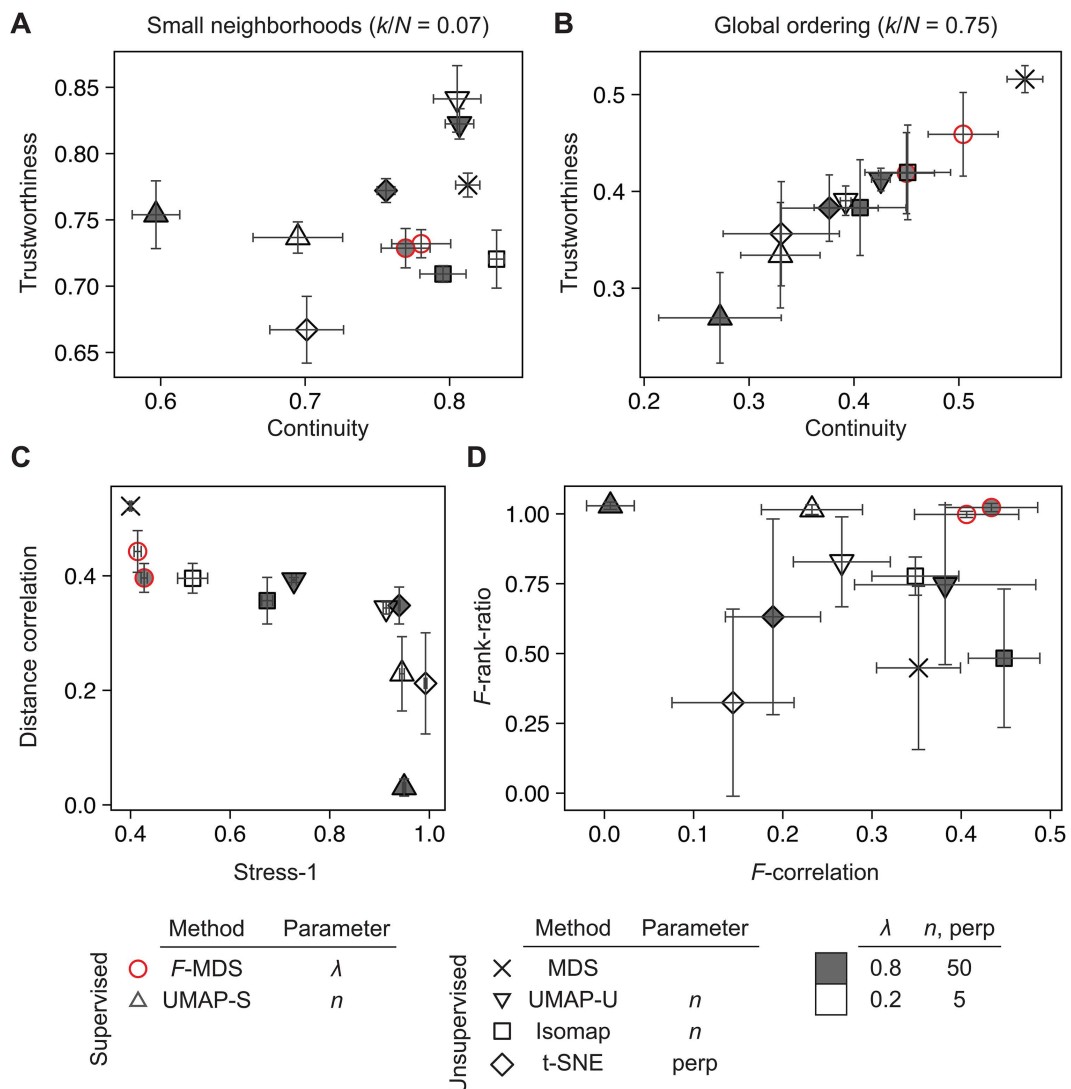

**Fig 4. Different quality metrics confirm consistent preservation of the semisynthetic data pattern with *F*-informed MDS.** Six dimension reduction methods were evaluated to test their preservation of (A) local and (B) global structure by calculating trustworthiness and continuity using two nearest-neighbor numbers, $k=14$ (local), $k=150$ (global). The methods were also evaluated using (C) global distortion metrics (Stress-1 and Shepard diagram correlation) and (D) statistical inference metrics, including the ratio in statistical significance (*F*-rank-ratio) and correlation in *F*-ratios (*F*-correlation) using a randomly permuted label set. The following hyperparameters were applied to each method: $\lambda$ (*F*-MDS), number of neighbors $n$ (UMAP, supervised (-S) and unsupervised (-U)), perplexity "perp" (t-SNE), and the number of shortest dissimilarities $n$ (Isomap). Three semisynthetic datasets of $N=200$ were generated as described in section "Semisynthetic data" of Methods. The standard deviations were calculated and displayed with error bars.

and S5B) or distance-based metrics such as pairwise distance correlation (Shepard diagram) or Stress-1. The latter two metrics revealed that graph-based methods such as UMAP and t-SNE consistently resulted in higher Stress-1 values than MDS methods (Figs 4C and S5C). This finding confirms that *F*-MDS does not significantly alter the global data structure compared to metric MDS. Finally, applying *F*-based metrics to evaluate these ordination methods showed their varying capabilities in representing distance structure in a lower dimension (Figs 4D and S5D). Methods that incorporate group labels, such as *F*-, superMDS and supervised UMAP (UMAP-S), had *F*-rank-ratio values close to unity, indicating that group differences were accurately preserved in their 2D representations. In contrast, *F*-correlation was higher in

unsupervised methods and those designed to retain distance structures, such as metric MDS, Isomap, and $F$-MDS, consistent with earlier observations from pairwise distance correlation.

**Algal-associated bacterial communities.** We next analyzed a real microbial community as another example to verify the performance of these ordination methods. Each sample presented a compositional structure of relative abundances based on 16S rRNA gene expression, identifying 72 bacterial taxa collected from mesocosms of the alga *P. tricornutum* [68,69]. Thirty-six community samples, half of which were co-cultured with the host *P. tricornutum* and the other half without, were retrieved and re-analyzed from previous work [54] (section Methods "Real microbiome community"). This dataset was chosen for our main analysis and evaluation because metric MDS did not detect a group difference ($p_z \approx 0.5$), whereas their original structures were statistically more different ($p_x < 0.1$). Other benchmark datasets, e.g., human gut microbiome, did not require additional iterations for $F$-MDS, as metric MDS already showed the significant differences ($p_z = 0.004$, cirrhosis; 0.004, type 2 diabetes). For all datasets, weighted Unifrac [17] was chosen as the distance metric to obtain the distance structure. Six quality metrics were again used to compare across seven dimension reduction methods, including one based on self-supervised learning (SimCLR [70]), to further explore its capability of arranging the algal microbiome data for interpretation (Appendix E, S1 File).

Similarly, UMAP-U and Isomap achieved the highest scores for preserving local neighborhoods, slightly outperforming $F$-MDS as measured by trustworthiness. In contrast, other supervised or label-incorporating methods had continuity scores that were 22–44% lower than Isomap (Fig 5A). For global structure preservation, however, $F$-MDS, metric MDS and Isomap were the top performers based on the two metrics as well as the pairwise distance correlation (Figs 5B, 5C, and S6). Finally, statistical evaluations showed that $F$-MDS was again effective in retaining both statistical significance and $F$-distributions in its lower dimensional representation (Figs 5D and S7). While SimCLR had an $F$-rank-ratio close to unity, comparable to other supervised ordinations – indicating high classification accuracy for clustering these data – its preservation scores for local and global structures were the lowest among all ordinations. Taken together, these multiple evaluations suggest that $F$-MDS is best suited for representing ecological data in terms of global ordering and group-based patterns, while its local structure representation is comparable to that of UMAP.

### $F$-MDS adjusts group clusters to visualize statistical differences

After evaluating $F$-MDS using different quality metrics, we applied the method to understand how sample groups were differentiated in 2D representations. The semi-synthetic and bacterial community datasets described earlier were used for visualization, noting that their group differences were not distinguishable under 2D representations with metric MDS ($p_z \geq 0.5$), whereas in the original dimension, they are statistically more distinct ($p_x < 0.1$). To detect structural changes in these representations, visualizations were generated using metric MDS and $F$-MDS with varying hyperparameters, and the results were compared by calculating the inter-centroid distance and variance of each group.

For the simulated dataset, we observed a shift in group centroids when hyperparameter $\lambda$ increased from zero to one, with a unidirectional shift that separated the groups (Fig 6). The increase in inter-group centroid distances was more pronounced at lower hyperparameter values ($\lambda < 0.2$), with an increase rate 1.9–25.7 fold higher than those at higher $\lambda$ values (S8 Fig). The finding suggests that the re-arrangement of the MDS representation, constrained by the majorization, slowed down and stabilized at high $\lambda$. This is consistent with our earlier iteration number analysis (Fig 2), where the computation epoch number reached a minimum, satisfying the objective $p_z \approx p_x$ as the hyperparameter was reached unity. Additionally, we observed a steady decrease in each group's variance by $8.5 \pm 7.8\%$ as $\lambda$ increased from 0 and 1 (S8 Fig), as measured from the eigendecomposition of their covariances. This indicates that $F$-MDS further differentiated the groups by reducing their respective variances.

Similar trends were observed in the bacterial community dataset, where inter-group centroid distances were greater with non-zero hyperparameters than with metric MDS (Fig 6B). However, no statistical significance was observed in the correlation due to variability ($p = 0.521$, Spearman correlation, S8D Fig). The primary (long-axis) variances of each 2D

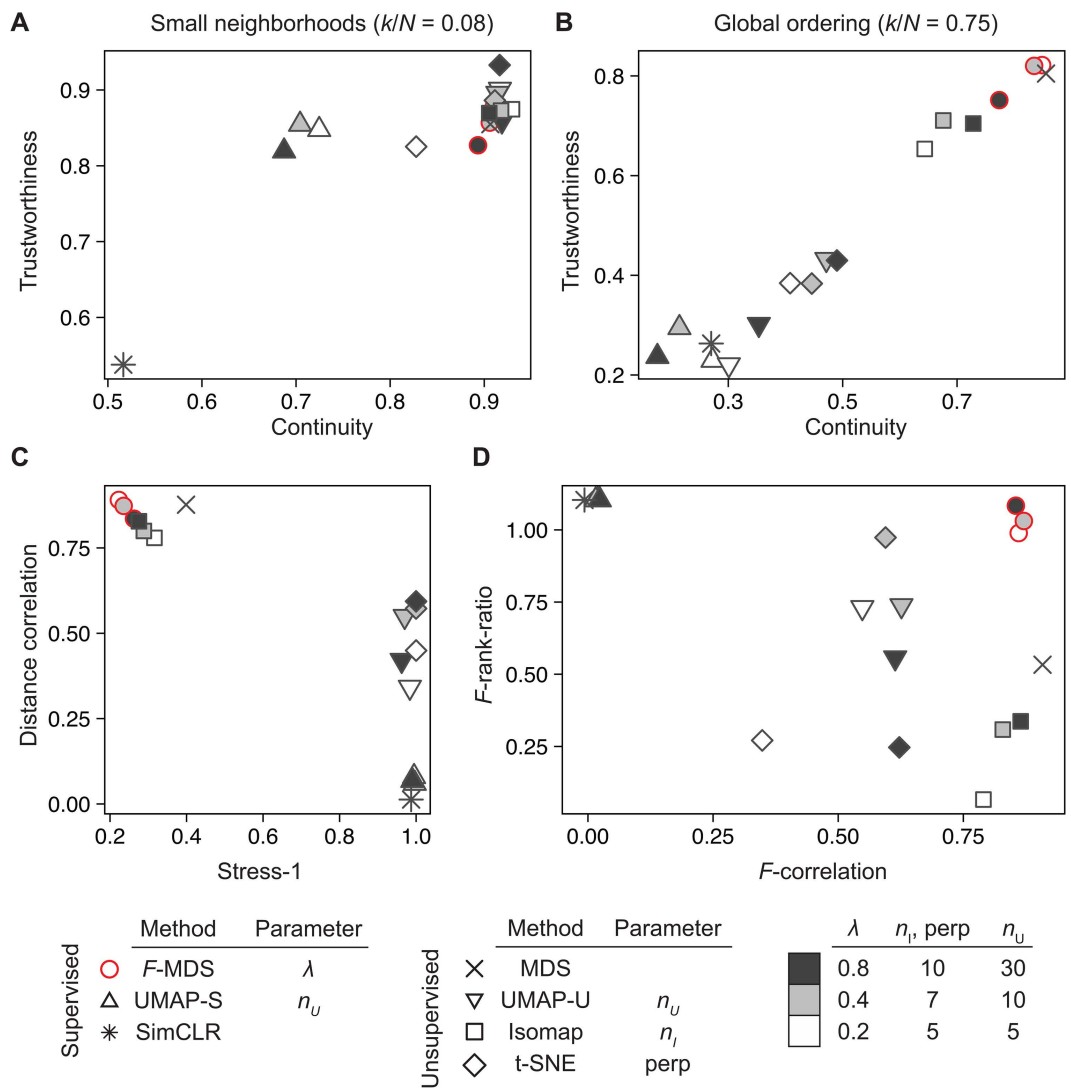

**Fig 5. Different quality metrics confirm consistent preservation of the algal microbiome pattern with *F*-informed MDS.** Seven dimension reduction methods were evaluated for their preservation of (A) local and (B) global structure by calculating trustworthiness and continuity using two nearest neighbor numbers $k=3$ (local), $k=27$ (global). The methods were also assessed based on (C) global distortion metrics (Stress-1 and Shepard plot correlation) and (D) the ratio of *p*-values (*F*-rank-ratio) and correlation in *F*-ratios (*F*-correlation) using randomly permuted label set. A dataset of $N=36$ bacterial communities was analyzed as described in section "Real microbiome community" of Methods. The following hyperparameters were applied: $\lambda$ (*F*-MDS), number of neighbors $n_U$ (UMAP, supervised (-S) and unsupervised (-U)), perplexity "perp" (t-SNE), and the number of shortest dissimilarities $n_I$ (Isomap).

group representation also consistently decreased by 24% and 18% per $\lambda$ for each group ($y=0$, 1 respectively, S8E Fig), while the decrease was less evident in the secondary (short-axis) variances (S8F Fig). Further comparisons across the 2D representations with different $\lambda$ values showed that the sample distributions remained similar, suggesting that *F*-MDS still did not significantly alter the metric MDS configuration.

**Differentiation of multi-group clusters.** To verify whether *F*-MDS can be applied to dataset consisting of more than two groups, we additionally considered a ternary dataset where each sample group followed a different zero-truncated log-normal distribution (Eq 11, S9 Fig). Similar to the earlier binary simulated dataset,

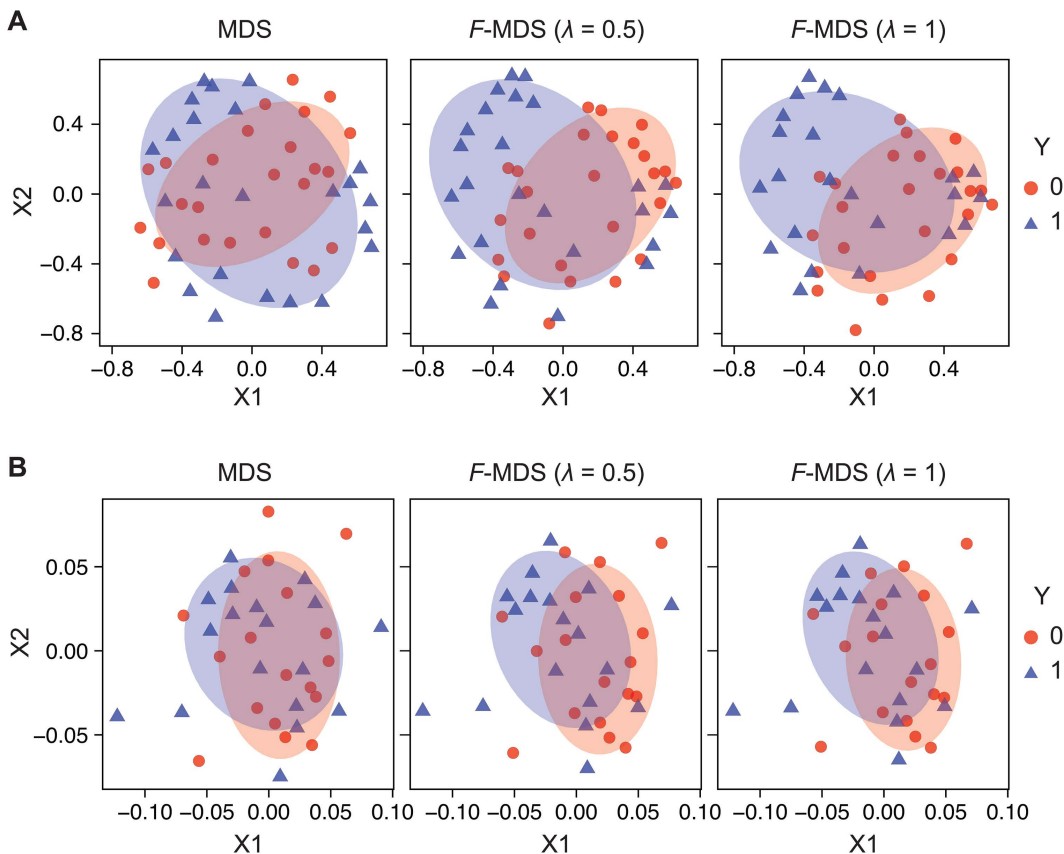

**Fig 6. *F*-informed MDS can visualize group-based data discrimination.** Two-dimensional representations of (A) a simulated dataset and (B) an algal microbiome dataset, generated using metric MDS and *F*-informed MDS (*F*-MDS) with two hyperparameter settings ($\lambda = 0.5, 1$). Ellipses of respective colors are drawn with a confidence level of 0.68.

the distribution was elliptically shaped, resulting in an indistinguishable representation in metric MDS ($p_z = 0.965$), whereas the original structures are statistically different ($p_x = 0.053$). Computing *F*-MDS with a hyperparameter $\lambda = 0.5$ showed that the sample groups deviated from each other, making the differences more visible when ellipses with a 68% confidence level were drawn (Fig 7). While this deviation led to a statistically significant 2D representation ($p_z = 0.001$), the ternary class setting rendered the difference less pronounced compared to binary data representations.

## Discussion

In this work, we proposed a weakly supervised multidimensional scaling method based on the *F*-ratio and characterized its representations by varying hyperparameters and the epoch number of the majorization algorithm. Evaluations using simulated and bacterial community datasets showed that our *F*-informed MDS outperformed existing MDS-based and other dimension reduction methods in preserving local and global structures, as well as class-based information (e.g., statistical inference). The datasets used in this study represented cases where metric MDS failed to identify group differences through visualizations, partly due to the loss of such information during dimensionality reduction. Our dispersion-based approach expanded the applicability of MDS by introducing an additional constraint to minimally adjust the representation while directly enabling group-associated inferences.

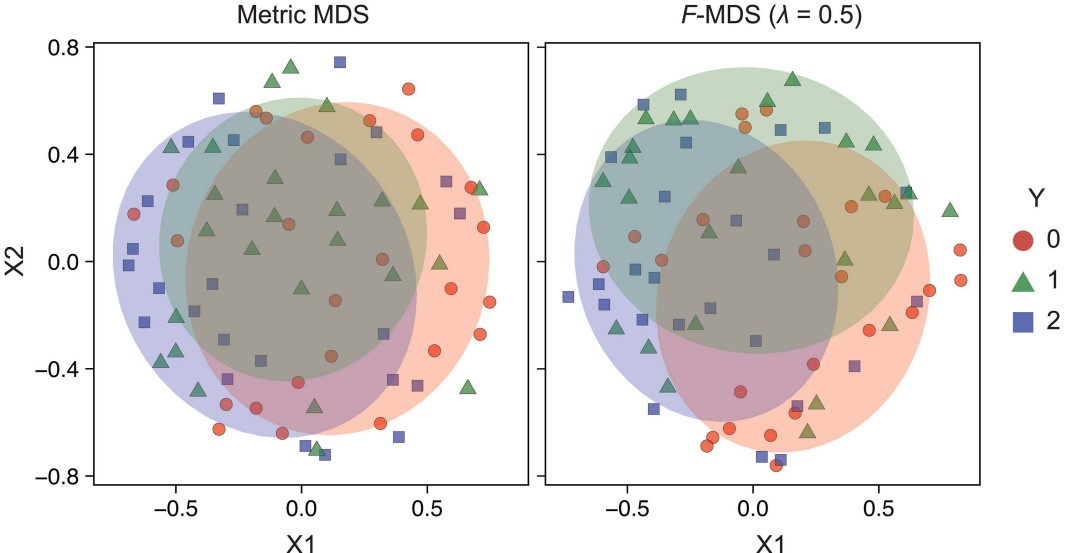

**Fig 7. Multi-class data is discriminated based on statistical inference.** Comparison of metric MDS and *F*-informed MDS ($\lambda = 0.5$) visualizations using a four-dimensional, three-class simulated dataset. Each dataset group follows normal distribution with the same covariance but different means (S9 Fig). Ellipses of respective colors are drawn with a confidence level of 0.68. Semisynthetic dataset of $N = 75$ was generated (see the section "Semisynthetic data" of the Methods.).

In microbiome research, for example, applying *F*-informed MDS to bacterial community datasets can allow users to interpret biologically meaningful patterns with higher confidence. It is also possible that the method could be employed in other domains where multivariate signals are processed based on distance metrics, such as metabolomics data clustering [71], network connectivity analysis with spatial neuroimaging [72], or spectral analysis for pharmaceutics quality control [73].

One limitation of *F*-informed MDS, however, is that its computation relies on the initial configuration from metric MDS, and the computational cost increase with iterations of majorization and label permutations (a summary of method costs is provided in Appendix F, S1 File). Indeed, majorization for MDS has been known for its high computational cost [48], although its numerical convergence is guaranteed [74] and we did not observe any stability issues in our simulation. Recent theoretical advances in computational algorithms, including majorization [75] or (stochastic) gradient descent [39,40] open doors for improving our framework toward unrestricted initialization and more efficient iterations. Additionally, empirical observations suggest that *F*-MDS representations became less reliant on the choice of hyperparameter at higher values. Once the relationship between the two objective-constituting terms is clarified, it could lead to a viable strategy for selecting an optimal hyperparameter without requiring recursive procedures such as cross-validation.

In summary, the proposed *F*-MDS provides a useful tool for visualizing high-throughput microbiome data while simultaneously delivering statistical testing results. We anticipate that this method will be beneficial for broader applications in microbiology and ecology data analysis. The R implementation of *F*-MDS is available at https://bioconductor.org/packages/FinfoMDS/.

## Supporting information

**S1 File. Supplementary Appendices.**
(PDF)

**S1 Fig. Visualization of a semisynthetic dataset using SparseDOSSA.** (A) Heatmap of its log-scaled relative abundance by its microbial taxa and sample number. (B) Density histogram of the data and (C) principal coordinates analysis (PCoA) and PERMANOVA $p$-values based on PCoA results with Euclidean distance ($p_z$) and original structure with Bray-Curtis dissimilarity ($p_x$). $N = 200$ samples were generated.
(PDF)

**S2 Fig. Stabiliity and trajectories of PERMANOVA $p$-values from $F$-MDS representations.** (A) Trajectories of $p_z$ across 50 epochs for $\lambda = 0.2, 0.5$, and $0.8$. Red dashed lines indicate the region of validity ($|p_z - p_x| < 0.05$). Pink points highlight where the stopping rule triggers, capturing the optimal statistical alignment before further oscillations occur. (B) Trajectories of $p$-values with the stopping rule ($|p_z - p_x| < 0.05$). Heatmaps show $p_z$ values plotted against training epochs for hyperparameters $\lambda \in [0, 1]$. Triplicate datasets of varying sizes ($N = 50, 100, 200, 500$) were evaluated to demonstrate consistent stabilization across conditions.
(PDF)

**S3 Fig. Pairwise distance analyses from $F$-informed MDS and supervised MDS using semisynthetic data.** The plots are titled with the dataset size $N$ and compared across different methods with hyperparameter values as follows: $\lambda$, $F$-MDS; $\alpha$, superMDS. After calculating pairwise distances with Bray-Curtis dissimilarity (original) and Euclidean (2D representation), (B) their Pearson correlation coefficient and (C) normalized stress (Stress-1) were obtained. Error bars are standard deviation of triplicates.
(PDF)

**S4 Fig. Hyperparameter selection using grid search.** An objective function $f_{obj}(\lambda)$ was defined to simultaneously reflect the number of training epochs and the preservation of the original structure (see Equation S17). (A) For each semisynthetic dataset size $N$, $f_{obj}(\lambda)$ is plotted against the hyperparameter $\lambda$. The minimum value of $f_{obj}(\lambda)$ is highlighted in red. (B) The optimal hyperparameter $\lambda_{min}$ is plotted against dataset size $N$, with each dataset represented by a symbol.
(PDF)

**S5 Fig. Comparison of quality metrics of benchmark ordination methods.** Trustworthiness and continuity to evaluate (A) Local structural preservation is assessed using trustworthiness and continuity. (B) Global structural preservation is similarly evaluated with trustworthiness and continuity. (C) Global distortion is quantified by Stress-1 and Pearson correlation of Shepard diagrams. (D) Preservation of statistical inference is measured by the $F$-rank-ratio and $F$-correlation using randomly permuted label sets. The following hyperparameters were used for each method: $\lambda$ for $F$-MDS, number of neighbors $n$ for UMAP (both supervised (-S) and unsupervised (-U)), perplexity (perp) for t-SNE, and the number of shortest dissimilarities $n$ for Isomap. Error bars represent the standard deviation across triplicate measurements.
(PDF)

**S6 Fig. Shepard plot from algal microbiome dataset with eight ordination methods.** The plots are titled with the respective method and hyperparameter values as follows: $\lambda$, $F$-MDS; $\alpha$, superMDS; Nearest neighbors number, supervised (-S) or unsupervised (-U) UMAP; Perplexity, t-SNE; Shortest dissimilarities number, Isomap; none, neural network (NN). X- and Y-axis denote distances in the original and embedding dimensions, respectively.
(PDF)

**S7 Fig. $F$-correlation plot from algal microbiome dataset.** Pseudo $F$-ratios comparing the original dimension (x-axis) and from eight dimension reduction methods (y-axis) with algal microbiome data. Pseudo $F$'s were calculated by randomly permuting labels by 500 times. Highlighted with red denotes the location of $F$'s from unpermuted labels. Each plot is titled with the method and hyperparameter used.
(PDF)

**S8 Fig. Cluster centroid and variances of *F*-MDS representations.** Cluster centroids and variances of *F*-MDS representations are shown for semisynthetic datasets of size $N = 50$ (A–C), $N = 100$ (D–F), $N = 200$ (G–I), $N = 500$ (J–L) and for the algal microbiome (M–O). The first column in each row displays the distance between group centroids. The second and third columns show the variance of each group, measured along the long and short principal axes, respectively. For the variance panels, blue and red colors denote groups 1 and 2. Error bars represent the standard deviation across triplicate measurements.
(PDF)

**S9 Fig. Visualization of a ternary semisynthetic dataset.** (A) Heatmap of its log-scaled relative abundance by its microbial taxa and sample number. (B) Density histogram of the data and (C) principal coordinates analysis (PCoA) and PERMANOVA *p*-values based on PCoA results with Euclidean distance ($p_z$) and original structure with Bray-Curtis dissimilarity ($p_x$). $N = 75$ semisynthetic data were generated.
(PDF)

## Acknowledgments

The authors acknowledge the MIT Office of Research Computing and Data for providing high performance computing resources that have contributed to the research results reported within this paper. We thank C. Belthangady and J.-Y. Lee for guidance in developing neural network models, and T. Sapsis and R.D. Braatz for their advice.

## Author contributions

**Conceptualization:** Hyungseok Kim, Soobin Kim.

**Data curation:** Hyungseok Kim, Soobin Kim, Jeffrey A. Kimbrel.

**Formal analysis:** Hyungseok Kim, Soobin Kim.

**Funding acquisition:** Xavier Mayali, Cullen R. Buie.

**Investigation:** Hyungseok Kim, Soobin Kim, Jeffrey A. Kimbrel.

**Methodology:** Hyungseok Kim.

**Project administration:** Xavier Mayali, Cullen R. Buie.

**Software:** Soobin Kim.

**Supervision:** Megan M. Morris, Xavier Mayali, Cullen R. Buie.

**Validation:** Hyungseok Kim, Soobin Kim.

**Visualization:** Hyungseok Kim.

**Writing – original draft:** Hyungseok Kim, Soobin Kim.

**Writing – review & editing:** Hyungseok Kim, Soobin Kim, Jeffrey A. Kimbrel, Megan M. Morris, Xavier Mayali, Cullen R. Buie.

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
