## [Decision Letter · Decision Letter 0]

6 Jun 2025

Multidimensional scaling informed by F-statistic: Visualizing grouped microbiome data with inference

PLOS Computational Biology

Dear Dr. Buie,

Thank you for submitting your manuscript to PLOS Computational Biology. After careful consideration, we feel that it has merit but does not fully meet PLOS Computational Biology's publication criteria as it currently stands. Therefore, we invite you to submit a revised version of the manuscript that addresses the points raised during the review process.

Please submit your revised manuscript within 60 days Aug 06 2025 11:59PM. If you will need more time than this to complete your revisions, please reply to this message or contact the journal office at ploscompbiol@plos.org. Please include the following items when submitting your revised manuscript:

We look forward to receiving your revised manuscript.

Kind regards,

Tao Wang

Academic Editor

PLOS Computational Biology

Zhaolei Zhang

Section Editor

PLOS Computational Biology

**Additional Editor Comments:**

The manuscript was reviewed by two referees. One reviewer recommends rejection, citing substantial concerns regarding the contribution and novelty of the work. The other acknowledges some merits but recommends major revisions. Notably, both reviewers find the simulation results unconvincing and insufficient to support the authors’ claims. Given these concerns, I do not believe the manuscript currently meets the standards of PLOS Computational Biology. The authors will need to carefully and comprehensively address these issues—some of which are fundamental—should they wish to resubmit.

**Journal Requirements:**

3) Please ensure that the funders and grant numbers match between the Financial Disclosure field and the Funding Information tab in your submission form. Note that the funders must be provided in the same order in both places as well.

**Reviewers' comments:**

Reviewer's Responses to Questions

**Comments to the Authors:**

Reviewer #1: Yes, the review is uploaded as an attachment

Reviewer #2: In this study, Kim and colleagues introduce a multidimensional scaling (MDS) method based on an F-statistic for grouped microbiome data. However, I think the authors overstate the importance of this method and the potential of using this method for analyzing and interpreting microbiome data.

The proposed approach is designed to preserve the hypothesis testing output for group differences between the original dataset and the MDS representation. However, the authors do not provide any explanations or reasoning (Lines 107–152) regarding why the significance of group differences between the original dataset and the MDS representation should be similar. For example, if group differences are significant in the original dataset but not in the MDS representation, does this imply that the MDS representation is not reliable? A significant group difference in the original dataset could be due to dimensions beyond the 2-dimensional MDS, which are not captured in the MDS representation.

A second major concern is the authors' claim that the proposed method is specifically designed for compositional data in microbiology and ecology. There is no clear connection between compositional data analysis and the proposed method since it only requires a distance matrix between samples.

A third major concern is that the simulation settings (Lines 154–168) are too simplistic to demonstrate the usefulness of MDS methods. In the simulation studies, the dimension of the original dataset is only 4, and the degrees of freedom are reduced to 3 due to the compositional constraint. Why not plot the scatter plots for all variable pairs in this case? Additionally, there appear to be typos in Equations (9) and (10) on Page 8. In Equation (9), "N" should be the variable dimension 4, and in Equation (10), "x_i" should be normalized to reflect the compositional nature of microbiome data, where relative abundances are measured and the count matrix from sequencing technologies should be normalized to sum to 1 in each sample.

Overall, I am unconvinced that this study provides enough importance and novelty to warrant publication in PLoS Computational Biology.

**Have the authors made all data and (if applicable) computational code underlying the findings in their manuscript fully available?**

The PLOS Data policy requires authors to make all data and code underlying the findings described in their manuscript fully available without restriction, with rare exception (please refer to the Data Availability Statement in the manuscript PDF file). The data and code should be provided as part of the manuscript or its supporting information, or deposited to a public repository. For example, in addition to summary statistics, the data points behind means, medians and variance measures should be available. If there are restrictions on publicly sharing data or code —e.g. participant privacy or use of data from a third party—those must be specified.requires authors to make all data and code underlying the findings described in their manuscript fully available without restriction, with rare exception (please refer to the Data Availability Statement in the manuscript PDF file). The data and code should be provided as part of the manuscript or its supporting information, or deposited to a public repository. For example, in addition to summary statistics, the data points behind means, medians and variance measures should be available. If there are restrictions on publicly sharing data or code —e.g. participant privacy or use of data from a third party—those must be specified.requires authors to make all data and code underlying the findings described in their manuscript fully available without restriction, with rare exception (please refer to the Data Availability Statement in the manuscript PDF file). The data and code should be provided as part of the manuscript or its supporting information, or deposited to a public repository. For example, in addition to summary statistics, the data points behind means, medians and variance measures should be available. If there are restrictions on publicly sharing data or code —e.g. participant privacy or use of data from a third party—those must be specified.requires authors to make all data and code underlying the findings described in their manuscript fully available without restriction, with rare exception (please refer to the Data Availability Statement in the manuscript PDF file). The data and code should be provided as part of the manuscript or its supporting information, or deposited to a public repository. For example, in addition to summary statistics, the data points behind means, medians and variance measures should be available. If there are restrictions on publicly sharing data or code —e.g. participant privacy or use of data from a third party—those must be specified.

Reviewer #1: Yes

Reviewer #2: None

PLOS authors have the option to publish the peer review history of their article (what does this mean?). If published, this will include your full peer review and any attached files.). If published, this will include your full peer review and any attached files.). If published, this will include your full peer review and any attached files.). If published, this will include your full peer review and any attached files.

...

Reviewer #1: No

Reviewer #2: No

**Figure resubmission:**

**Reproducibility:**



---

## [Decision Letter · Decision Letter 1]

31 Oct 2025

Multidimensional scaling informed by F-statistic: Visualizing grouped microbiome data with inference

PLOS Computational Biology

Dear Dr. Buie,

Thank you for submitting your manuscript to PLOS Computational Biology. After careful consideration, we feel that it has merit but does not fully meet PLOS Computational Biology's publication criteria as it currently stands. Therefore, we invite you to submit a revised version of the manuscript that addresses the points raised during the review process.

Please submit your revised manuscript within 60 days Dec 31 2025 11:59PM. If you will need more time than this to complete your revisions, please reply to this message or contact the journal office at ploscompbiol@plos.org. Please include the following items when submitting your revised manuscript:

We look forward to receiving your revised manuscript.

Kind regards,

Tao Wang

Academic Editor

PLOS Computational Biology

Zhaolei Zhang

Section Editor

PLOS Computational Biology

**Additional Editor Comments:**

While one reviewer is satisfied with the revised manuscript, the other still has several major concerns that need to be addressed.

**Reviewers' comments:**

Reviewer's Responses to Questions

**Comments to the Authors:**

Reviewer #1: The authors have addressed the previous concerns very well.

Reviewer #3: Thank you for your careful revisions. However, I still have several major concerns:

1. You defined the regression function f_z to link F_x^{\Pi} and F_z^{\Pi}. However, in Algorithm 1, the data pair used is (F_x^{\Pi_1}, F_z^{\Pi_2}), meaning that F_x and F_z are computed from different permutations. In that case, why should a reasonable regression function f_z exist between them? For instance, it would be analogous to expecting a regression relationship between a sample u \sim N(0,1) and another independent sample v \sim N(1,1), which clearly have no meaningful correspondence.

2. (i) Is it correct to say that a smaller value of \lambda leads to better preservation of the original distance structure, whereas a larger \lambda places more emphasis on preserving group differences?

(ii) Equation (S17) is used as the objective function to select the optimal hyperparameter \lambda. However, it is rather unconventional to incorporate computation time as an explicit component of the loss function. Could you clarify the underlying principle or rationale for determining the optimal value of \lambda?

(iii) For a fixed \lambda, the stopping criterion of the algorithm is not clearly specified. As suggested in Fig 1B, the procedure terminates when p_x is not substantially smaller than p_z. Could you clarify the exact rule—for instance, |p_z - p_x| / p_x < eps? Furthermore, if the algorithm were allowed to continue, would p_z continue to decrease toward zero, fluctuate, or converge to p_x? This behavior is not clearly reflected in the reported results (e.g., Figs. 2 and S2).

(iv) This question is related to (i), (ii) and (iii). Is it correct to say that, for a given \lambda, the algorithm either fails to converge or that p_z always converges to p_x? Furthermore, is there a threshold value of \lambda such that, for values below this threshold, the algorithm does not converge, while for values above it, p_z converges to p_x? If so, would it be reasonable to select the optimal \lambda as this threshold?

3. What is the real-world application of the proposed F-MDS? Demonstrating a concrete application would make the importance of your method more convincing, particularly since it prioritizes preserving group differences even at the expense of some loss in the original distance structure.

4. Line 184: “with all but one feature mean differently adjusted for each dataset” seems unclear. It appears that you intend to say that only one feature is set to have different means between the two groups, but the current wording implies that all features except one have different means. In addition, why did you choose to set only one differential feature? This seems too few given that there are 332 features in total.

5. It is unclear which results (tables and figures) correspond to which datasets, as you introduce multiple semi-synthetic and real datasets in the “Dataset and Evaluation” section, but these datasets are not explicitly referenced in the Results section.

**Have the authors made all data and (if applicable) computational code underlying the findings in their manuscript fully available?**

The PLOS Data policy requires authors to make all data and code underlying the findings described in their manuscript fully available without restriction, with rare exception (please refer to the Data Availability Statement in the manuscript PDF file). The data and code should be provided as part of the manuscript or its supporting information, or deposited to a public repository. For example, in addition to summary statistics, the data points behind means, medians and variance measures should be available. If there are restrictions on publicly sharing data or code —e.g. participant privacy or use of data from a third party—those must be specified.requires authors to make all data and code underlying the findings described in their manuscript fully available without restriction, with rare exception (please refer to the Data Availability Statement in the manuscript PDF file). The data and code should be provided as part of the manuscript or its supporting information, or deposited to a public repository. For example, in addition to summary statistics, the data points behind means, medians and variance measures should be available. If there are restrictions on publicly sharing data or code —e.g. participant privacy or use of data from a third party—those must be specified.requires authors to make all data and code underlying the findings described in their manuscript fully available without restriction, with rare exception (please refer to the Data Availability Statement in the manuscript PDF file). The data and code should be provided as part of the manuscript or its supporting information, or deposited to a public repository. For example, in addition to summary statistics, the data points behind means, medians and variance measures should be available. If there are restrictions on publicly sharing data or code —e.g. participant privacy or use of data from a third party—those must be specified.requires authors to make all data and code underlying the findings described in their manuscript fully available without restriction, with rare exception (please refer to the Data Availability Statement in the manuscript PDF file). The data and code should be provided as part of the manuscript or its supporting information, or deposited to a public repository. For example, in addition to summary statistics, the data points behind means, medians and variance measures should be available. If there are restrictions on publicly sharing data or code —e.g. participant privacy or use of data from a third party—those must be specified.

Reviewer #1: Yes

Reviewer #3: None

PLOS authors have the option to publish the peer review history of their article (what does this mean?). If published, this will include your full peer review and any attached files.). If published, this will include your full peer review and any attached files.). If published, this will include your full peer review and any attached files.). If published, this will include your full peer review and any attached files.

...

Reviewer #1: No

Reviewer #3: No

**Figure resubmission:**

**Reproducibility:**



---

## [Decision Letter · Decision Letter 2]

22 Jan 2026

Multidimensional scaling informed by F-statistic: Visualizing grouped microbiome data with inference

PLOS Computational Biology

Dear Dr. Buie,

Thank you for submitting your manuscript to PLOS Computational Biology. After careful consideration, we feel that it has merit but does not fully meet PLOS Computational Biology's publication criteria as it currently stands. Therefore, we invite you to submit a revised version of the manuscript that addresses the points raised during the review process.

We look forward to receiving your revised manuscript.

Kind regards,

Tao Wang

Academic Editor

PLOS Computational Biology

Zhaolei Zhang

Section Editor

PLOS Computational Biology

**Additional Editor Comments:**

Unfortunately, one referee remains dissatisfied with the revision and raises serious concerns regarding the rationale for ordering the F-values and the apparent lack of convergence of the algorithm, as evidenced by the fluctuations shown in Figure 2. Please take these comments very seriously.

At this point, I cannot predict the final outcome. I am willing to consider another revision that directly and convincingly addresses the latest referee reports. After reviewing your revision and responses, I will decide whether to send the manuscript for another round of review or return it to you with a rejection if the issues appear unlikely to be resolved.

**Journal Requirements:**

**Reviewers' comments:**

Reviewer's Responses to Questions

**Comments to the Authors:**

Reviewer #3: 1. I do not understand why it is reasonable to order the F-values. Consider the extreme case in which the original F_z and F_x are perfectly negatively linearly correlated, which appears to be favored by your method since it yields a perfectly linear relationship. Is this the intended outcome ?

Alternatively, consider this from another perspective. Since independent permutations can be applied to z and x, F_z and F_x may be independent. Suppose there are two representations with the same raw stress. In this case, how does the confirmatory term help to reveal group differences and decide which representation is better?

2. The algorithm does not appear to converge. In Figure 2 of your response, p_z exhibits substantial and seemingly random fluctuations.

**Have the authors made all data and (if applicable) computational code underlying the findings in their manuscript fully available?**

The PLOS Data policy requires authors to make all data and code underlying the findings described in their manuscript fully available without restriction, with rare exception (please refer to the Data Availability Statement in the manuscript PDF file). The data and code should be provided as part of the manuscript or its supporting information, or deposited to a public repository. For example, in addition to summary statistics, the data points behind means, medians and variance measures should be available. If there are restrictions on publicly sharing data or code —e.g. participant privacy or use of data from a third party—those must be specified.requires authors to make all data and code underlying the findings described in their manuscript fully available without restriction, with rare exception (please refer to the Data Availability Statement in the manuscript PDF file). The data and code should be provided as part of the manuscript or its supporting information, or deposited to a public repository. For example, in addition to summary statistics, the data points behind means, medians and variance measures should be available. If there are restrictions on publicly sharing data or code —e.g. participant privacy or use of data from a third party—those must be specified.requires authors to make all data and code underlying the findings described in their manuscript fully available without restriction, with rare exception (please refer to the Data Availability Statement in the manuscript PDF file). The data and code should be provided as part of the manuscript or its supporting information, or deposited to a public repository. For example, in addition to summary statistics, the data points behind means, medians and variance measures should be available. If there are restrictions on publicly sharing data or code —e.g. participant privacy or use of data from a third party—those must be specified.requires authors to make all data and code underlying the findings described in their manuscript fully available without restriction, with rare exception (please refer to the Data Availability Statement in the manuscript PDF file). The data and code should be provided as part of the manuscript or its supporting information, or deposited to a public repository. For example, in addition to summary statistics, the data points behind means, medians and variance measures should be available. If there are restrictions on publicly sharing data or code —e.g. participant privacy or use of data from a third party—those must be specified.

Reviewer #3: None

PLOS authors have the option to publish the peer review history of their article (what does this mean?). If published, this will include your full peer review and any attached files.). If published, this will include your full peer review and any attached files.). If published, this will include your full peer review and any attached files.). If published, this will include your full peer review and any attached files.

...

Reviewer #3: No

**Figure resubmission:**
---

## [Editor Report · Decision Letter 3]

8 Mar 2026

Dear Professor Buie,

We are pleased to inform you that your manuscript 'Multidimensional scaling informed by F-statistic: Visualizing grouped microbiome data with inference' has been provisionally accepted for publication in PLOS Computational Biology.

Best regards,

Tao Wang

Academic Editor

PLOS Computational Biology

Zhaolei Zhang

Section Editor

PLOS Computational Biology

---

## [Editor Report · Acceptance letter]

PCOMPBIOL-D-25-00623R3

Multidimensional scaling informed by F-statistic: Visualizing grouped microbiome data with inference

Dear Dr Buie,

I am pleased to inform you that your manuscript has been formally accepted for publication in PLOS Computational Biology. Your manuscript is now with our production department and you will be notified of the publication date in due course.

With kind regards,

Anita Estes
